# A Training-Free Debiasing Framework with Counterfactual Reasoning for Conversational Emotion Detection

**Geng Tu**[1,2] *, **Ran Jing**[1,2] *, **Bin Liang**[3], **Min Yang**[4], **Kam-Fai Wong**[3], **Ruifeng Xu**[1,2,5] †

[1]Harbin Institute of Technology, Shenzhen, China
[2]Guangdong Provincial Key Laboratory of Novel Security Intelligence Technologies
[3]The Chinese University of Hong Kong, Hong Kong, China
[4]SIAT, Chinese Academy of Sciences, Shenzhen, China
[5]Peng Cheng Laboratory, Shenzhen, China
tugeng0313@gmail.com, xuruifeng@hit.edu.cn

## Abstract

Unintended dataset biases typically exist in existing Emotion Recognition in Conversations (ERC) datasets, including label bias, where models favor the majority class due to imbalanced training data, as well as the speaker and neutral word bias, where models make unfair predictions because of excessive correlations between specific neutral words or speakers and classes. However, previous studies in ERC generally focus on capturing context-sensitive and speaker-sensitive dependencies, ignoring the unintended dataset biases of data, which hampers the generalization and fairness in ERC. To address this issue, we propose a Training-Free Debiasing framework (TFD) that operates during prediction without additional training. To ensure compatibility with various ERC models, it does not balance data or modify the model structure. Instead, TFD extracts biases from the model by generating counterfactual utterances and contexts and mitigates them using simple yet empirically robust element-wise subtraction operations. Extensive experiments on three public datasets demonstrate that TFD effectively improves generalization ability and fairness across different ERC models[1].

## 1 Introduction

Emotion recognition in conversations (ERC) has garnered substantial research interest in recent years (Mao et al., 2021; Xie et al., 2021; Jiang et al., 2022; Tu et al., 2023b). This attention stems from its promising applications in various domains, including recommendation systems and dialogue generation (Tu et al., 2022a). ERC aims to identify the emotion of each utterance in conversations. Current efforts primarily focus on modeling context- and speaker-sensitive dependencies (Lian et al., 2021). This includes recurrent-based network (Hazarika

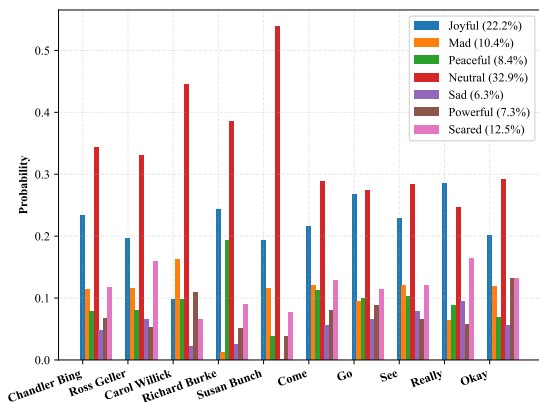

Figure 1: The probability distribution of different speakers and neutral words, confirms various biases in ERC.

et al., 2018; Majumder et al., 2019; Ghosal et al., 2020; Jiao et al., 2020; Li et al., 2022), transformer-based network (Zhong et al., 2019; Lian et al., 2021; Shen et al., 2021a; Ong et al., 2022), and graph-based network (Ghosal et al., 2019; Shen et al., 2021b; Saxena et al., 2022).

Although these endeavors have achieved satisfactory results, they all overlook unintended biases present in the datasets, thereby impeding their generalization capabilities and fairness (Goyal et al., 2017; Niu et al., 2021). Different from other tasks, in ERC, these biases may manifest at both the utterance level, such as label and speaker biases, as well as the word level, like the neutral word bias. **Label bias**, as depicted in Fig. 1, is evident in the EmoryNLP dataset's training set, where 32.9% (2485 utterances) are labeled as 'neutral' and 6.3% (474 utterances) are labeled as 'sad'. Numerous prior studies (Dixon et al., 2018; Zhang et al., 2020) have indicated that models trained on such imbalanced data are susceptible to the tendency of predominantly predicting the majority class. **Speaker and neutral word biases** arise from trained models exhibiting strong associations between specific words and particular emotion categories. For instance, in EmoryNLP, it is highly probable for

---

* The first two authors contribute equally to this work.
† Corresponding author.

[1]The code is available at https://github.com/TuGengs/TFD.

Richard Burke to be associated with emotions such as 'joyful', 'neutral', and 'peaceful', rather than 'mad' as depicted in Figure 1. Additionally, neutral words also display such biases in emotion distribution. Consequently, models tend to unfairly assign utterances containing these emotionless keywords to specific categories based on biased statistical information, rather than relying on intrinsic textual semantics (Qian et al., 2021; Liu and Avci, 2019; Waseem and Hovy, 2016).

Existing debiasing methods mainly involve two approaches: Data-level manipulations, like resampling (Qian et al., 2020) and generating counterfactual samples (Wang and Culotta, 2021), aim to balance the training set but increase training time. Model-level balancing mechanisms, such as counterfactual reasoning (Tian et al., 2022) and reweighting (Zhang et al., 2020), adjust category influence during training but require careful strategy selection and retraining from scratch.

In contrast, our proposed Training-Free Debiasing framework (TFD) focuses on counterfactual reasoning during the prediction process without extra data augmentation and any training costs. Because generating counterfactual examples while maintaining conversation structure integrity is challenging, inspired by Qian et al. (2021), we employ a masking mechanism to obtain counterfactual examples from the original input data to extract biases suffered from trained models. We then effectively mitigate these extracted biases using empirically robust element-wise subtraction operations. Our main contributions are as follows:

- To our knowledge, we are the first debiasing study for ERC.

- We propose a TFD to tackle biases in ERC without extra data augmentation and any training costs, which employs counterfactual utterances and contexts to extract these biases suffered from the trained model.

- We conduct extensive experiments on three public datasets to illustrate the effectiveness of TFD in enhancing generalization and fairness across various ERC models. And the TFD-based baseline demonstrates the ability to outperform state-of-the-art methods.

## 2 Related Work

**ERC:** The emotion generation theory (Gross and Barrett, 2011) emphasizes the importance of con-

textual information for identifying emotions. RNN-based models have been commonly used to capture context dependencies (Poria et al., 2017), but they struggle to distinguish between historical utterances (Lian et al., 2021). In order to overcome this drawback, researchers have directed their attention toward memory networks (Jiao et al., 2020; Hazarika et al., 2018). Additionally, the role of participants in emotional response classification (ERC) is crucial (Wen et al., 2023), leading to the development of speaker-specific (Kim and Vossen, 2021; Majumder et al., 2019), graph-based models (Ghosal et al., 2019; Tu et al., 2022b; Shen et al., 2021b) and so on. However, these approaches still lack commonsense knowledge, which is important for human-like performance (Tu et al., 2023a). To tackle this, researchers have integrated external knowledge sources such as COMET (Bosselut et al., 2019), SenticNet (Cambria et al., 2022) and ConceptNet (Speer et al., 2017) into their models (Zhong et al., 2019; Ghosal et al., 2020; Zhao et al., 2022). Despite these advancements, unintended dataset biases have been neglected, which hinders the generalization of ERC models.

**Dataset debiasing:** To address dataset bias, one approach is to manipulate the data during model training to prevent unintended biases from being captured. These manipulations include techniques like data balance or resampling methods mentioned in studies such as (Geng et al., 2007; Kang et al., 2016; Sun et al., 2018; Wang and Culotta, 2021), as well as data augmentation methods described by Qian et al. (2020). Another common strategy is to incorporate model-level balancing mechanisms. These mechanisms aim to mitigate bias by using unbiased embeddings (Kaneko and Bollegala, 2019), adjusting thresholds (Kang et al., 2019), and applying instance weighting techniques (Tian et al., 2022; Zhao et al., 2017; Zhang et al., 2020). However, the data-level strategy results in additional manual costs for data manipulations and longer training time. On the other hand, the model-level strategy necessitates a meticulous selection of balancing techniques and retraining whenever there are changes in the balancing mechanisms.

## 3 Methodology

### 3.1 Task Definition

Consider a conversation $\mathbf{C} = \{\mathbf{u}_1, \mathbf{u}_2, ..., \mathbf{u}_n\}$ consisting of $n$ utterances. Each utterance $\mathbf{u}_i$ is spoken by one of the speakers in $\mathbf{S} = \{\mathbf{s}_1, \mathbf{s}_2, ..., \mathbf{s}_m\}$.

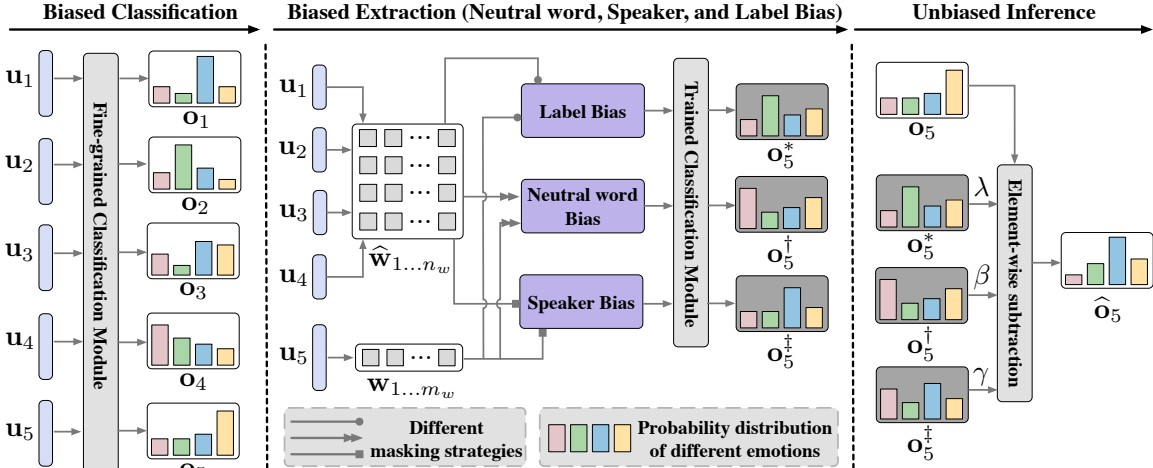

Figure 2: The proposed TFD framework. The bias extraction and unbiased inference are only conducted during the testing phase. Mathematical symbols in the illustration are consistent with the formulas in the paper.

The objective of the ERC task is to predict the emotion label $\mathbf{y}_i$ for each utterance $u_i$ by incorporating the preceding $\omega$ utterances $\{\mathbf{u}_{i-\omega}, ..., \mathbf{u}_{i-2}, \mathbf{u}_{i-1}\}$. $\mathbf{Y} = \{\mathbf{y}_1, \mathbf{y}_2, ..., \mathbf{y}_n\}$ represents a predefined set of emotions.

## 3.2 Overview

Considering the additional manual and training costs brought by data-level and model-level debiasing approaches. we propose TFD, a Training-Free Debiasing framework, which draws inspiration from the effective use of counterfactual reasoning to address biases in computer vision, as demonstrated in prior works (Tang et al., 2020; Yang et al., 2021; Niu et al., 2021). TFD aims to mitigate biases in trained models through counterfactual reasoning in prediction. It can be integrated into various ERC models, and we employ Roberta as the classification model to exemplify its implementation. TFD comprises three primary components: biased classification, bias extraction, and unbiased inference (refer to Fig. 2).

## 3.3 Biased Classification

**Input Format:** Utterance representation $\mathbf{x}_i$ is typically generated by directly inputting query utterance $\mathbf{u}_i$ into a pre-trained language model (PLM). Considering that concatenating utterances alone may not significantly improve results as PLMs lack dialogue structure during pretraining. Therefore, inspired by Kim and Vossen (2021), we integrate dialogue structure and contextual information ex-

plicitly into the input text.

$$\mathbf{x}_i = \{\mathbf{X}_c(\mathbf{u}_i)\} \tag{1}$$
$$\widehat{\mathbf{x}}_i = \{\mathbf{X}_h(\mathbf{u}_{i-\omega}), ..., \mathbf{X}_h(\mathbf{u}_{i-1})\} \tag{2}$$

where $\omega$ represents the size of the context, $\mathbf{X}_h$ and $\mathbf{X}_c$ represent the encoding strategies for historical utterances and the current utterance, respectively, without using future information. For $\mathbf{X}_c$, we prepend *speaker says:* at the start of historical and current utterances to indicate the speaker's information. Additionally, $\langle s \rangle$ and $\langle /s \rangle$ are employed to enclose each utterance for emphasis. In essence, the current utterance can be represented as follows:

$$\mathbf{X}_c(u_i) = \langle s \rangle \, \mathbf{s}_i \, says : \mathbf{u}_i \, \langle /s \rangle \tag{3}$$

For $\mathbf{X}_h$, *speaker says:* serves as the sole token, considering the support of contexts and speakers.

$$\mathbf{X}_h(u_j) = \mathbf{s}_j \, says : \mathbf{u}_j, \, \forall j \in (i - \omega, i) \tag{4}$$

**Classification Module:** Traditional fine-tuning methods often use the class token for classification because most tasks involve plain text without specific structures. However, in our case, the input text $\mathbf{x}_i$ has a principal-subordinate structure, where the query utterance takes the leading position and the contexts play a supporting role. Moreover, there is a temporal structure present. Therefore, we propose a fine-tuning classification module that considers the characteristics of ERC. The module's specifics will be outlined below.

$$\mathbf{E}_c = \mathbf{mean}\left(\mathbf{PLM}\left(\mathbf{x}_i\right)\right) \tag{5}$$
$$\mathbf{E}_h = \mathbf{mean}\left(\mathbf{PLM}\left(\widehat{\mathbf{x}}_i\right)\right) \tag{6}$$
$$\mathbf{o}_i = \mathbf{MLP}\left(\mathbf{FC}(\mathbf{E}_c \oplus \mathbf{E}_h)\right) \tag{7}$$

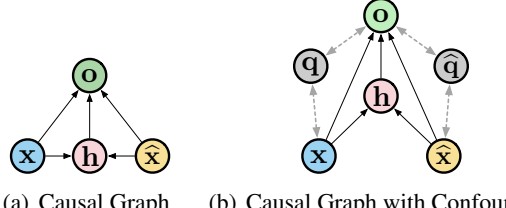

(a) Causal Graph    (b) Causal Graph with Confounder

Figure 3: The comparison between factual ERC and counterfactual ERC.

where $\mathbf{o}_i$ is the prediction emotion. $\mathbf{E}_{c/h}$ is the features of $\mathbf{u}_i$ and its contexts, generated by PLM. $\oplus$ denotes a concatenation operation. $\mathbf{FC}$ is the fully connected layer with $\mathbf{Tanh}$ activation function. $\mathbf{MLP}$ represents the multilayer perceptron.

### 3.4 Bias extraction

#### 3.4.1 Causal Graph

Causal graph (Glymour et al., 2016) is a comprehensive representation of the causal relationships among variables. It consists of nodes (variables) and directed edges (arrows) that indicate causality. In this directed acyclic Bayesian graphical model, denoted as $\mathcal{G} = \{\mathcal{N}, \mathcal{E}\}$, the nodes represent variables, and the arrows represent causal relationships between them. For instance, $\mathbf{x} \rightarrow \mathbf{Y}$ indicates that $\mathbf{x}$ is the cause and $\mathbf{Y}$ is the effect, meaning that the value of $\mathbf{Y}$ is influenced by $\mathbf{x}$.

We define the causal graph of ERC in Fig. 3.(a). Nodes $\mathbf{x}$, $\widehat{\mathbf{x}}$ and $\mathbf{h}$ represent the input utterance, its context, and the combined feature of $\mathbf{x}$ and $\widehat{\mathbf{x}}$ respectively. The final predictive logits $\mathbf{o}$ take inputs from the three branches: the direct effect of the input $\mathbf{x}$ and $\widehat{\mathbf{x}}$ on $\mathbf{o}$ via $\mathbf{x} \rightarrow \mathbf{o}$ and $\widehat{\mathbf{x}} \rightarrow \mathbf{o}$, as well as the indirect effect of the input $\mathbf{x}$ and $\widehat{\mathbf{x}}$ on $\mathbf{o}$ via the combined feature $\mathbf{h}$, i.e. $\mathbf{h} \rightarrow \mathbf{o}$.

#### 3.4.2 Speaker and Neutral Word Biases

As shown in Fig. 3.(b), there exists unobserved confounders $\mathbf{q}$ and $\widehat{\mathbf{q}}$, which is the cause of the spurious correlation between the $\mathbf{x}$, $\widehat{\mathbf{x}}$ and $\mathbf{o}$. Such unobserved confounders may occur due to trained models exhibiting strong associations between specific words (such as neutral word and speaker) and particular emotion categories. As a result, models tend to unfairly assign outputs containing these keywords to specific categories based on biased statistical information, more than releasing on introductory textual semantics, which is actually not reasonable for the ERC task. To decouple the spurious correlation, we use the backdoor adjustments (Gly-

mour et al., 2016) with do-calculus to calculate the corresponding intervention distribution:

$$\mathbf{P}(\mathbf{o}|\mathbf{do}(\mathbf{x}, \widehat{\mathbf{x}})) = \mathbf{P}(\mathbf{o}|\mathbf{do}(\mathbf{x} = \mathbf{c}, \widehat{\mathbf{x}} = \widehat{\mathbf{c}})) \quad (8)$$

where $\mathbf{c}$ and $\widehat{\mathbf{c}}$ can take any form as long as they are no longer influenced by $\mathbf{q}$, effectively breaking the connection between $\mathbf{x}$, $\widehat{\mathbf{x}}$ and $\mathbf{q}$. To extract neutral word or speaker biases, in the causal intervention operation, $\mathbf{c}$ / $\widehat{\mathbf{c}}$ are set to $\mathbf{c}^\dagger$ / $\widehat{\mathbf{c}}^\dagger$ or $\mathbf{c}^\ddagger$ / $\widehat{\mathbf{c}}^\ddagger$, which aims to retain only neutral words or speaker-related words. By revealing specific words tied to the emotion categories, the goal is to expose any spurious connections in the trained model and emphasize their potential negative impact.

$$\mathbf{c}_i^\ddagger = \langle s \rangle \{\mathbf{w}_1, \langle mask \rangle, ..., \mathbf{w}_{n_w}\} \langle /s \rangle \\ \forall \mathbf{w}_i \leftarrow \langle mask \rangle \notin \mathbf{S} \quad (9)$$

$$\widehat{\mathbf{c}}_i^\dagger = \{\widehat{\mathbf{w}}_1, ..., \widehat{\mathbf{w}}_{m_w}\}, \forall \widehat{\mathbf{w}}_i \leftarrow \langle mask \rangle \notin \mathbf{S} \quad (10)$$

where $\mathbf{S}$ denotes the set of speakers. In this case, since the model cannot see any word except for words related to speakers in $\mathbf{x}$ and $\widehat{\mathbf{x}}$ after masking, the final predictive logits $\mathbf{o}_i^\ddagger$ reflect the pure influence of speakers to the trained biased model. Similarly, we can also obtain the counterfactual utterance and the counterfactual historical utterance $\widehat{\mathbf{c}}_i^\dagger$ and $\widehat{\mathbf{c}}_i^\dagger$ by using pysentiment library[2] to determine non-neutral words for masking.

#### 3.4.3 Label Bias

To address the bias of models to favor the majority class in imbalanced data, we generate counterfactual utterances $\mathbf{c}$ and contexts $\widehat{\mathbf{c}}$ through causal intervention to extract label biases.

$$\mathbf{c}_i^* = \langle s \rangle \{\mathbf{w}_1, \langle mask \rangle, ..., \mathbf{w}_{n_w}\} \langle /s \rangle \quad (11)$$

$$\widehat{\mathbf{c}}_i^* = \{\widehat{\mathbf{w}}_1, ..., \widehat{\mathbf{w}}_{m_w}\}, \forall \mathbf{w}_i, \widehat{\mathbf{w}}_i \leftarrow \langle mask \rangle \quad (12)$$

where $\langle mask \rangle$ is used to hide a single token in the input sequence $\mathbf{x}$. The corresponding predictive logits $\mathbf{o}_i^*$ only reflect the impact of the label bias because the model cannot observe any words after the complete masking.

### 3.5 Unbiased Inference

Debiasing predictive logits $\mathbf{o}_i$ from the biased trained model can be formulated using a simple element-wise subtraction operation.

$$\widehat{\mathbf{o}}_i = \mathbf{o}_i - \lambda \mathbf{o}_i^* - \beta \mathbf{o}_i^\dagger - \gamma \mathbf{o}_i^\ddagger \quad (13)$$

---

[2]Available at https://pypi.python.org/pypi/pysentiment

| Dataset | Dialogues | | | Utterances | | |
|---|---|---|---|---|---|---|
| | train | val | test | train | val | test |
| IEMOCAP | 120 | | 31 | 5,810 | | 1,623 |
| MELD | 1039 | 114 | 280 | 9,989 | 1,109 | 2610 |
| EmoryNLP | 659 | 89 | 79 | 7,551 | 954 | 984 |

| Dataset | Classes | Metric |
|---|---|---|
| IEMOCAP | 6 | Weighted Avg. F1 |
| MELD | 3 and 7 | Weighted Avg. F1 over 3 and 7 classes |
| EmoryNLP | 3 and 7 | Weighted Avg. F1 over 3 and 7 classes |

Table 1: Statistics of experimental datasets.

where $\lambda$, $\beta$, and $\gamma$ are three independent parameters. It's important to note that biases have varying impacts on the final classification and are not equally important. Thus, we use elastic scaling to determine three adjustable scaling factors that optimize the model's performance on the validation set.

# 4 Experiments

## 4.1 Datasets

We conduct experiments on three datasets: IEMO-CAP (Busso et al., 2008), EmoryNLP (Zahiri and Choi, 2018), and MELD (Poria et al., 2019a). The statistics of datasets are shown in Table 1.

**IEMOCAP** consists of dyadic sessions where actors perform improvisations or scripted scenarios. Each utterance is labeled with one of the emotions: angry, happy, sad, neutral, excited, or frustrated. Since there is no dedicated validation set in this dataset, we follow the approach from Shen et al. (2021b) by using the last 20 dialogues from the training set for validation.

**MELD** is a multi-party conversation dataset collected from the TV show *Friends*. Each utterance is annotated with one of the emotions: fear, surprise, anger, disgust, sadness, neutral, or joy, and one of the sentiments: neutral, negative, or positive.

**EmoryNLP** consists of multi-party sessions from the TV show *Friends* and each utterance is labeled with one emotion from the set: joyful, scared, peaceful, sad, powerful, mad, or neutral, along with one sentiment, as suggested in Ghosal et al. (2020), from the set: neutral, negative, or positive.

## 4.2 Comparison Models

We compare our proposed framework with various ERC baselines, including RNN-based models: CauAIN (Zhao et al., 2022), COSMIC (Ghosal et al., 2020), DialogueRNN (Majumder et al., 2019); Memory networks: CoMPM (Lee and Lee, 2022), AGHMN (Jiao et al., 2020); Graph-based models: DAG-ERC (Shen et al., 2021b), SKAIG (Li et al., 2021); Transformer-based models: KET (Zhong et al., 2019), BERT_BASE (Kenton and Toutanova, 2019), EmoBERTa (Kim and Vossen, 2021), Roberta (Liu et al., 2019); Generative models: Curie (Olmo et al., 2021), ChatGPT (Ouyang et al., 2022)[3].

## 4.3 Experimental Settings

All models mentioned in Table 4 have released their source codes, and we have used the same settings as the original papers. The Roberta-based method (Baseline) described in this paper is based on EmoBERTa (Kim and Vossen, 2021), with the difference being that we do not consider the impact of future utterances on the model. Furthermore, we have utilized a grid search technique to determine the optimal values for the parameters $\lambda$, $\beta$, and $\gamma$ on the validation set. The grid search is performed with a step size of 0.1 and a range spanning from -2 to 2. The results reported in the tables are the average scores from 5 random runs on the test set.

# 5 Experimental Results

## 5.1 Main Results

As shown in Table 2, fine-tuning at the utterance level alone is unsatisfactory in ERC because it relies on context and the speaker's state information. EmoBERTa, a redesigned Roberta model with adjusted input structures, exhibits significant performance improvement. CoMPM surprisingly outperforms other methods in IEMOCAP and MELD datasets, showcasing the effectiveness of complex memory mechanisms. Graph-based models generally surpass RNN-based models in IEMOCAP and EmoryNLP datasets, capturing local context well in lengthy conversations. However, in MELD, where data is from TV shows, the coherence between consecutive utterances may be lacking, reducing the advantage of graph-based models. Large models underperform in ERC, possibly due to limitations in capturing intricate interactions, especially in lengthy conversations, as evidenced by their failure on the IEMOCAP dataset. Furthermore, ChatGPT's understanding of emotions is not limited to predefined categories, which may affect its emotional comprehension. Prompt tuning the Curie model improves performance but is not sufficient.

---

[3]Please refer to the appendix for the prompt templates used for ChatGPT and Curie.

| Methods | IEMOCAP (6-cls) | MELD (3-cls) | MELD (7-cls) | EmoryNLP (3-cls) | EmoryNLP (7-cls) |
|---|---|---|---|---|---|
| ChatGPT[§] | 40.07 | 60.07 | 54.37 | 51.93 | 37.55 |
| Curie[§] | 57.33 | - | 65.01 | - | 37.40 |
| BERT_BASE[♯] | 61.19 | - | 56.21 | - | 33.15 |
| RoBERT[♯] | 54.55 | 72.12 | 62.02 | 55.28 | 37.29 |
| EmoBERTa[♭] | 68.57 | - | 65.61 | - | - |
| DialogueRNN[♭] | 61.21 | 66.10 | 56.27 | 48.93 | 31.70 |
| KET[♭] | 59.56 | - | 58.18 | - | 34.39 |
| AGHMN[♭] | 62.70 | - | 58.10 | - | - |
| DAG-ERC[♭] | 68.03 | - | 63.65 | - | 39.02 |
| SKAIG[♭] | 66.96 | - | 65.18 | - | 38.88 |
| COSMIC[♭] | 65.28 | 73.20 | 65.21 | 56.51 | 38.11 |
| CauAIN[♭] | 67.61 | - | 65.46 | - | - |
| CoMPM[♭] | 69.46 | - | **66.52** | - | 38.93 |
| Baseline[§] | 69.18 | 72.57 | 65.00 | 56.84 | 38.81 |
| Baseline[§] + TFD | **70.43** (↑ *1.25%*) | **73.72** (↑ *1.15%*) | 66.19 (↑ *1.19%*) | **58.28** (↑ *1.41%*) | **40.51** (↑ *1.70%*) |
| **w/o** Label debiasing | 69.66 (↓ *0.77%*) | 72.61 (↓ *1.11%*) | 65.35 (↓ *0.94%*) | 57.05 (↓ *1.23%*) | 39.19 (↓ *1.32%*) |
| **w/o** Neutral word debiasing | 69.45 (↓ *0.98%*) | 72.77 (↓ *0.95%*) | 65.31 (↓ *0.88%*) | 57.27 (↓ *1.01%*) | 39.28 (↓ *1.23%*) |
| **w/o** Speaker debiasing | 69.81 (↓ *0.62%*) | 72.78 (↓ *0.94%*) | 65.28 (↓ *0.91%*) | 57.33 (↓ *0.95%*) | 39.48 (↓ *1.03%*) |

Table 2: Weighted Avg. F1 score (%) of different methods. Best test scores are in bold. [§] represents our re-implementation results. Results with [♯] and [♮] are respectively retrieved from Zhong et al. (2019) and Ghosal et al. (2020). Results with [♭] are retrieved from the original papers.

This does not imply that generative methods are ineffective for ERC; more task-specific factors like imbalanced samples and long-term context modeling require careful consideration.

Considering that most methods rely on using Roberta for utterance representation extraction, we use Roberta as the underlying model to validate the effectiveness of TFD. We can observe that the performance on sentiment and emotion identification of our proposed TFD-based method is significantly boosted. And it achieves the best improvement result of F1 on the EmoryNLP dataset, i.e., 1.70%. And the Baseline[§]+TFD outperforms all compared methods on different datasets, except for slightly underperforming the CoMPM model on the MELD dataset. The failure could possibly be attributed to the elimination of the influence of future utterances on our model, as evidenced by the performance gap between our Baseline[§] and EmoBERTa.

## 5.2 Ablation Studies

To investigate the impact of each component of our proposed method, we conduct an ablation study on Baseline[§] + TFD, and the results are shown in Table 2. **w/o** represents the removing operation. The results suggest that all components of the TFD framework have worked and all the improvements are statistically significant, as evidenced by the paired t-test results with a p-value < 0.05.

### 5.2.1 Label Debiasing

Due to the tendency of models to prioritize the majority class, such as the emotions 'joyful', 'scared', and 'neutral', imbalanced data often leads to in-

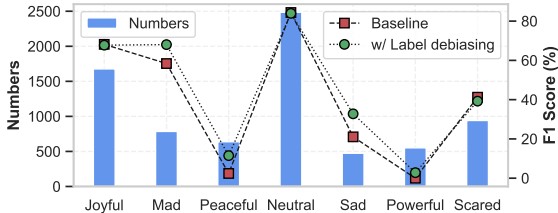

Figure 4: Performance of Baseline[§] on the EmoryNLP validation set across different categories.

adequate recognition performance for utterances belonging to the minority classes, as depicted in Fig. 4. Fortunately, with the introduction of the label debiasing strategy, this phenomenon has been effectively mitigated. From Fig. 4, we find that the debiased model significantly improves on the other minority classes, such as 'peaceful', 'sad', and 'powerful'. Despite a marginal decline in performance for the majority classes, the model exhibits overall improvement, with an increase from 39.19 to 40.51. The advantages outweigh the drawbacks in this case, which suggests the effectiveness of label debiasing and the reported results presented in Table 2 further validate this assertion.

### 5.2.2 Speaker and Neutral Word Debiasing

We have observed improvements in speaker and neutral word debiasing across various datasets. Specifically, when we applied the debiasing strategy of removing the speaker and neutral words to the EmoryNLP dataset, we noticed a significant decrease in the model's performance, resulting in a respective drop of 1.03% and 1.23%. However,

| Methods | IEMOCAP (6-cls) | MELD (3-cls) | MELD (7-cls) | EmoryNLP (3-cls) | EmoryNLP (7-cls) |
|---|---|---|---|---|---|
| Baseline[§] | 2.11 | 4.42 | 9.16 | 9.87 | 26.66 |
| w/ Label debiasing | 2.07 (↓ *0.04%*) | 4.16 (↓ *0.26%*) | 7.66 (↓ *1.50%*) | 8.34 (↓ *1.53%*) | 22.73 (↓ *3.93%*) |
| w/ Neutral word debiasing | 1.97 (↓ *0.14%*) | 4.05 (↓ *0.37%*) | 7.55 (↓ *1.61%*) | 8.54 (↓ *1.33%*) | 24.32 (↓ *2.34%*) |
| w/ Speaker debiasing | 2.08 (↓ *0.03%*) | 4.12 (↓ *0.30%*) | 7.10 (↓ *2.56%*) | 8.29 (↓ *1.58%*) | 23.42 (↓ *3.24%*) |

Table 3: Imbalance divergence or unfairness (%) of our underlying model (lower is better) for bias analysis.

the speaker debiasing approach does not yield the expected results on the IEMOCAP dataset, as illustrated in Table 2. This disparity can be explained by the distinction between the IEMOCAP dataset and the other two datasets obtained from TV shows, where each speaker possesses distinct personality traits and emotional tendencies. In contrast, the improvisations in IEMOCAP do not prioritize the shaping of character images, naturally reducing the inclination towards specific emotional expressions. In neutral word debiasing, the effectiveness varies to some degree, but overall, the debiasing process demonstrates relatively consistent improvement across various datasets due to the prevalent presence of neutral words in conversations.

### 5.3 Hyper-parameter Analysis

We also explore the effects of different coefficients on these debiasing strategies, as shown in Figure 5. Generally, they exhibit a trend of initially increasing and then decreasing, even falling below the Baseline[§] in the case of the neutral word debiasing. However, label debiasing is quite unique. After an initial increase, it consistently maintains stable performance, consistently outperforming the Baseline[§]. This reflects the severe impact of label bias on model performance, while the bias associated with neutral words is relatively mild.

### 5.4 Bias Analysis

Based on prior research (Xiang et al., 2020; Sweeney and Najafian, 2019; Qian et al., 2021), we use imbalance divergence as a metric to measure the degree of unfairness in prediction results. This metric allows us to assess how much a trained model's predictions favor specific predefined categories, quantifying the discriminatory nature of the model's behavior. A higher imbalance divergence in the model's predictions indicates more unfairness. As shown in Table 3, our TFD has all resulted in a decrease in the imbalance divergence of predicting outcomes. However, label debiasing does not achieve a significant decrease in performance on the IEMOCAP dataset compared to the other two datasets. This could be because although all

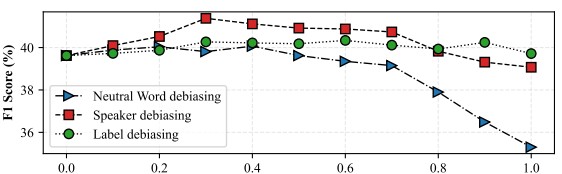

Figure 5: Performance of Baseline[§] with various debiasing strategies across different debiasing coefficients on the validation set of dataset EmoryNLP.

three datasets have class imbalance issues in the validation set, the severity is more pronounced in the other two datasets. In the IEMOCAP dataset, the ratio between the most frequent and least frequent classes in the training set is approximately 2.98:1, while in MELD and EmoryNLP, the ratios are 17.57:1 and 5.24:1, respectively. Additionally, for speaker debiasing, since IEMOCAP is derived from improvisational performances with less emphasis on character portrayal, the bias related to speakers is not as significant as in the other two datasets. The reported results in Table 2 also reflect this observation. However, neutral word debiasing shows a considerable decrease in performance across all three datasets due to the generality of neutral word debiasing.

### 5.5 Generalization Analysis

To evaluate the generalizability of our TFD framework, we conduct the experiment on various ERC models as shown in Table 4 and 5. With the exception of the imbalance divergence for AGHMN on the IEMOCAP dataset and DialogueRNN on the 3-class classification task on the MELD dataset, TFD has been effective in all other cases. Moreover, the F1 score for all methods has shown improvement, which demonstrates the effectiveness and generalization ability of our TFD framework.

### 5.6 Case Study

As shown in Table 6, in the first utterance, the speaker most likely expresses a neutral emotion, leading the model to associate her with a neutral emotion despite the phrase "Nice work." Removing the speaker bias results in the correct prediction. In

| Methods | IEMOCAP (6-cls) | MELD (3-cls) | MELD (7-cls) | EmoryNLP (3-cls) | EmoryNLP (7-cls) |
|---|---|---|---|---|---|
| DialogueRNN[§] | 66.79 | 72.46 | 64.65 | 55.80 | 38.34 |
| **w/ TFD** | 66.94 (↑ *0.15%*) | 73.17 (↑ *0.71%*) | 64.81 (↑ *0.16%*) | 56.69 (↑ *0.89%*) | 39.39 (↑ *1.05%*) |
| DAG-ERC[§] | 66.42 | 72.56 | 65.39 | 56.26 | 37.84 |
| **w/ TFD** | 66.57 (↑ *0.15%*) | 73.37 (↑ *0.81%*) | 65.53 (↑ *0.14%*) | 56.53 (↑ *0.27%*) | 38.59 (↑ *0.75%*) |
| AGHMN[§] | 65.21 | 72.45 | 64.19 | 54.61 | 37.86 |
| **w/ TFD** | 65.46 (↑ *0.25%*) | 72.92 (↑ *0.47%*) | 64.76 (↑ *0.57%*) | 56.09 (↑ *1.48%*) | 38.64 (↑ *0.78%*) |
| KET[§] | 63.16 | 71.42 | 62.67 | 54.62 | 37.73 |
| **w/ TFD** | 64.57 (↑ *1.41%*) | 72.31 (↑ *0.89%*) | 63.46 (↑ *0.79%*) | 56.89 (↑ *2.27%*) | 38.39 (↑ *0.66%*) |
| COSMIC[§] | 60.86 | 72.71 | 64.76 | 56.23 | 39.16 |
| **w/ TFD** | 61.50 (↑ *0.64%*) | 73.21 (↑ *0.50%*) | 64.99 (↑ *0.23%*) | 57.02 (↑ *0.79%*) | 39.85 (↑ *0.69%*) |

Table 4: Weighted Avg. F1 score (%) of different ERC methods for generalizability analysis.

| Methods | IEMOCAP (6-cls) | MELD (3-cls) | MELD (7-cls) | EmoryNLP (3-cls) | EmoryNLP (7-cls) |
|---|---|---|---|---|---|
| Baseline[§] | 2.11 | 4.42 | 9.16 | 9.87 | 26.66 |
| **w/ TFD** | 1.96 (↓ *0.15%*) | 4.02 (↓ *0.40%*) | 6.98 (↓ *2.18%*) | 7.84 (↓ *2.03%*) | 18.18 (↓ *8.48%*) |
| DialogueRNN[§] | 10.82 | 5.96 | 13.29 | 11.80 | 22.22 |
| **w/ TFD** | 8.09 (↓ *2.73%*) | 6.90 (↑ *0.94%*) | 10.26 (↓ *3.03%*) | 7.69 (↓ *4.11%*) | 8.29 (↓ *13.93%*) |
| DAG-ERC[§] | 17.46 | 7.44 | 17.80 | 11.78 | 21.41 |
| **w/ TFD** | 17.32 (↓ *0.14%*) | 7.29 (↓ *0.15%*) | 15.88 (↓ *0.15%*) | 9.68 (↓ *2.10%*) | 17.92 (↓ *3.49%*) |
| AGHMN[§] | 10.60 | 6.87 | 18.80 | 13.23 | 22.11 |
| **w/ TFD** | 10.60 (↓ *0.00%*) | 6.35 (↓ *0.52%*) | 16.99 (↓ *1.81%*) | 11.70 (↓ *1.53%*) | 16.71 (↓ *5.40%*) |
| KET[§] | 13.01 | 7.43 | 12.52 | 11.44 | 21.83 |
| **w/ TFD** | 9.58 (↓ *3.43%*) | 6.78 (↓ *0.65%*) | 10.06 (↓ *2.46%*) | 5.77 (↓ *5.67%*) | 20.55 (↓ *1.28%*) |
| COSMIC[§] | 11.60 | 13.72 | 25.38 | 13.21 | 24.20 |
| **w/ TFD** | 11.32 (↓ *0.28%*) | 11.19 (↓ *2.53%*) | 10.11 (↓ *15.27%*) | 10.66 (↓ *2.55%*) | 17.46 (↓ *6.74%*) |

Table 5: Imbalance divergence or unfairness (%) of different ERC methods for generalizability analysis.

| ID | Utterances for Prediction | **w/o** TFD | **w/** TFD | Golden Label |
|---|---|---|---|---|
| 1 | RACHEL: Okay get your coat! Oh! When did you unhook this? Nice work! | neural | joyful | joyful |
| 2 | CHANDLER: Act like a processor, people will think you're a processor. You're right. | joyful | peaceful | peaceful |
| 3 | RACHEL: Okay wait! | neural | anger | anger |

Table 6: Examples of utterances from the EmoryNLP dataset for the case study.

| Methods | IEMOCAP | MELD | EmoryNLP |
|---|---|---|---|
| Ours | 70.43 | 66.19 | 40.51 |
| **w/o** Emotion Shift | 74.76 | 75.40 | 52.99 |
| **w/** Emotion Shift | 65.02 | 59.52 | 36.24 |

Table 7: Analysis of TFD on Emotional Shift

the second utterance, because other neutral words are infrequent, except for pronouns, the neutral word "think" dominates the joyful emotion, which is rectified when the neutral word bias is eliminated. In the third utterance, the majority of samples indicate a neutral emotion. Although the context provides some clues, the model simply predicts the majority class. However, after removing the label bias, the model correctly predicts the emotion of the utterance based on its context.

### 5.7 Error Analysis

Most errors can be attributed to class imbalance, such as the low F1 score of 17.14 for the 'powerful' emotion in the EmoryNLP dataset. Our TFD may also make mistakes, as shown in Fig. 4, where certain samples of majority classes. This could

be because the coefficients used for the debiasing strategy are less detailed, as they are applied to the entire sample set instead of individual utterances or conversations. The latter approach would require significant tuning efforts. Furthermore, there are distribution differences between the validation and test sets, which can result in prediction errors for certain samples. Another concern is the issue of emotion shift (Poria et al., 2019b), where consecutive utterances express different emotions. This has been challenging for previous approaches as well. Table 7 shows that our TFD still struggles to effectively handle samples with emotional shifts compared to those without such shifts.

### 6 Conclusion

In this paper, we introduce TFD with counterfactual reasoning as a solution to address unintended dataset biases in ERC, which performs the debiasing operation only during the prediction phase without additional data augmentation or training expenses. Through extensive experiments on three public datasets, we demonstrate that TFD effec-

tively promotes the generalization capability and fairness of various ERC methods.

## Acknowledgements

We thank the anonymous reviewers for their valuable suggestions to improve the overall quality of this manuscript. This work was partially supported by the National Natural Science Foundation of China (62006062, 62176076), Natural Science Foundation of GuangDong 2023A1515012922, Shenzhen Foundational Research Funding JCYJ20210324115614039 and JCYJ20220818102415032, Guangdong Provincial Key Laboratory of Novel Security Intelligence Technologies 2022B1212010005.

## Limitations

When facing significant distribution differences between the validation and test sets, it becomes challenging to ensure the effectiveness of a method. In such cases, finding appropriate coefficients to address the biased prediction results becomes difficult. As shown in Fig. 5, while Label debiasing can always improve the model's performance, Neutral Word and Speaker debiasing are often unable to consistently guarantee effectiveness. Furthermore, label debiasing may not have an impact when dealing with samples with highly balanced classes, as label bias becomes negligible.

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

Figure 6: Prompt template for ChatGPT.

Figure 7: Prompt template for the Curie model.

## A   Prompt Templates

In this section, we primarily focus on explaining the methodology used to obtain the results for the ChatGPT and prompt-tuned Curie model in this paper. For ChatGPT, we employ a prompt template, as depicted in Fig. 6, to extract the emotion of each utterance in a conversation. This approach ensures the utilization of contextual information and facilitates the generation of well-formatted output results. On the other hand, for the Curie model, we directly access OpenAI's API for implementation. We utilize the prompt example presented in Fig. 7 to perform prompt-tuning, which enhances the model's performance in generating emotionally appropriate responses.