# OpenReview forum: "A Training-Free Debiasing Framework with Counterfactual Reasoning for Conversational Emotion Detection"
_EMNLP/2023/Conference — EMNLP 2023 Main_

### Official Review · Reviewer_p6EJ · 2023-08-04

**Soundness:** 4

**Excitement:**

4: Strong: This paper deepens the understanding of some phenomenon or lowers the barriers to an existing research direction.

**Paper Topic And Main Contributions:**

The paper introduces a new inference-time debiasing approach tailored to mitigate data collection and annotation biases in Emotion Recognition in Conversations (ERC). The proposed Training-free Debiasing Framework (TDF) defines causal graphs to model interactions between the input utterance, context, and emotion to discover biases and produce "biased" logits to subtract from the original logits. Experiments on three ERC datasets show that debiasing a pretrained RoBERTa with TDF leads to better emotion recognition and fairer prediction than a large set of baselines.

The paper has several strengths, including convincing results and ablation studies. Readability and structure, especially in section 3.4, can be improved by providing examples and extended descriptions.

**Questions For The Authors:**

MELD and EmoryNLP come from the same source and have similar statistics. What is the rationale for adding them both to the analysis?

What do you mean by "the replication model" in Table 2's caption? Also, in the same table, what is the baseline model? Is it RoBERTa base?

**Reasons To Accept:**

The paper's core idea is interesting in several aspects. It is efficient, as it does require fine-tuning. Moreover, bias detection and logit correction are general methodologies that can find use beyond ERC.

The authors compare the performance of TDF against many similar approaches and prove it better.

An ablation study proves that every correction helps improve the model.

**Reasons To Reject:**

The readability of Section 3 (and specifically 3.3 and 3.4) is the weakest point of the paper. The authors should expand it with some examples. The following is a list of less clear passages:
- Equation 2: are the ... standing for "concatenation"? Is the representation of the context built by concatenating the previous utterances? If so, do you use special tokens to separate utterances from the context?
- Is "speaker" a *variable* in the phrase "speaker say"? Because Eq. 3 and 4. suggest so.
- The authors state that "the \<mask\> token is utilized to highlight the emotional state of the sentence": what do you mean by that, and specifically "to highlight the emotional state"? This passage requires more clarity, perhaps achieved by providing an example.
- Why are you not using \<s\> and \</s\> in context utterances?
- L251 reports **h** is the variable representing "the combined features": what do you mean by that?
- Sections 3.4.2 and 3.4.3 are crucial to understanding the underlying methodology; however, the writing and notation take time to follow. I strongly recommend describing in detail and with an example of how neutral and speaker-related word sets are constructed.

**Reproducibility:**

3: Could reproduce the results with some difficulty. The settings of parameters are underspecified or subjectively determined; the training/evaluation data are not widely available.

**Reviewer Confidence:**

3: Pretty sure, but there's a chance I missed something. Although I have a good feel for this area in general, I did not carefully check the paper's details, e.g., the math, experimental design, or novelty.

**Typos Grammar Style And Presentation Improvements:**

Several citations need to be in the correct style. When used as subjects, for example, you should use \citet, e.g., "inspired by (Kim and Vossen, 2021)" should be "inspired by Kim and Vossen, 2021".

You have a mentioned dataset (L316), Dailydialog, that you later decided to remove from the paper.

---

> ### Author Rebuttal · Authors · 2023-08-28
>
> We would like to express our great appreciation for your valuable comments to improve the quality of this manuscript. We will release the source code after the acceptance of this paper to facilitate reproducibility by readers.
>
> **Answer of Reject#1:**
> &ensp;&ensp;Thanks for your insightful question. Yes, the context representation is built by concatenating each token in the historical utterances without using special tokens to separate these utterances, following the suggestion from the previous work (Kim and Vossen, 2021).
>
> *Kim, T., & Vossen, P. (2021). Emoberta: Speaker-aware emotion recognition in conversation with roberta. arXiv preprint arXiv:2108.12009.*
>
> **Answer of Reject#2:**
> &ensp;&ensp;Thanks for your valuable question. Yes, the term "speaker" appears as a variable in the phrase "speaker says.", evidenced by the indications in Equations 3 and 4. However, it is important to note that "speaker" may also appear as a variable in the utterance. We will detail it in the revision.
>
> **Answer of Reject#3:**
> &ensp;&ensp;Thanks for your valuable comment. In fact, there is no "<mask>" token. The correct Equation (3) is as follows:
> &ensp;&ensp;$X_{c}(u_{i})$ = <s\> ${s}_i$ says: ${u}_i$ </s>
> &ensp;&ensp;Inspired by prompts, we attempted to introduce the <mask> token to signify the emotional context of utterances. For instance, in analyzing the emotion in the utterance "Alice: I did well in the exam", we might employ the prompt "Alice felt <mask>". This <mask> token allows the Pre-trained Language Model (PLM) to function within a familiar framework, thereby enhancing its performance in downstream tasks. If the <mask> token is utilized, the input format of contexts in Equation (4) should be “$s_j$ <emo> says: $u_j$”, where <emo> represents the emotion label. However, during experimentation, this method did not yield the expected results. This could be attributed to the inference stage's challenge in obtaining the <emo> token within the input context format. If the model's predicted output is used as the <emo> token, incorrect predictions might mislead the emotion detection of the current utterance. Consequently, we decided to remove it. We sincerely apologize for this oversight. We have corrected it in the revision and will avoid these mistakes in the future.
>
> **Answer of Reject#4:**
> &ensp;&ensp;Thanks for your insightful question. The <s> and </s> are utilized to differentiate the current utterance from its context, i.e., the historical utterances.
>
> **Answer of Reject#5:**
> &ensp;&ensp;Thanks for your insightful question. The variable "h" symbolizes the combined representation of an utterance and its context, indirectly affecting the prediction outcomes in the causal graph.
>
> **Answer of Reject#6:**
> &ensp;&ensp;Thanks for your valuable comment. In the given utterance, *"CHANDLER says Act like a processor, people will think you’re a processor. You’re right."* with the emotion "peaceful". Our objective is to mitigate the impact of neutral words, like "think" which commonly appears in utterances with the emotion "joyful". Similarly, we aim to counter the influence of speakers, such as CHANDLER, who frequently convey "neutral" and "joyful" emotions. Therefore, we extract neutral word and speaker biases. Specifically, we mask tokens that do not belong to the set of speakers (CHANDLER is the only speaker in this example) to predict outcomes with speaker bias. We can also obtain prediction outcomes with neutral word biases by masking tokens that do not belong to the set of neutral words, such as "right". We will detail it in the revision.
>
> **Answer of Question#1:**
> &ensp;&ensp;Thanks for your insightful question. There is no particular rationale behind the selection of these two datasets, as they are widely utilized in ERC. This facilitates comparison with previous methods.
>
> **Answer of Question#2:**
> &ensp;&ensp;Thanks for your valuable question. The replication model refers to our re-implementation results. Specifically, the results for ChatGPT and Curie in Table 2 are obtained by using our prompt templates provided in the appendix. The Roberta-based method (Baseline) is based on EmoBERTa, with the difference being that we did not consider the impact of future utterances on the model. **Please refer to lines 351-355 on page 5.** This exclusion is due to the impracticality of utilizing future utterances to predict the current one’s emotions in real-world scenarios.
>
> **Answer of Typos Grammar Style And Presentation Improvements:**
> &ensp;&ensp;We greatly appreciate your input, and in response to your suggestions, we have made the necessary revisions to rectify the error. Your insightful comments are incredibly helpful in improving the quality of our work.

---

### Official Review · Reviewer_LsJC · 2023-08-11

**Typos Grammar Style And Presentation Improvements:** N/A
**Soundness:** 3

**Excitement:**

4: Strong: This paper deepens the understanding of some phenomenon or lowers the barriers to an existing research direction.

**Missing References:**

N/A

**Paper Topic And Main Contributions:**

The paper proposed a debiasing framework with counterfactual reasoning for conversational emotion detection. The main contributions of this paper are:
- They propose a TFD to tackle biases in REC without extra data augmentation and any training costs.
- They conducted extensive experiments on three public datasets to illustrate the effectiveness of TFD.

**Questions For The Authors:**

In Table 2, I found the results are not consistent with the three items (i.e., w/ label debiasing, w/ neutral word debiasing, and w/ speaker debiasing)

**Reasons To Accept:**

The paper proposed a new method TFD and achieve good results on three public datasets when compared to several SOTAs.

**Reasons To Reject:**

No specific reason to reject from my side.

**Reproducibility:**

3: Could reproduce the results with some difficulty. The settings of parameters are underspecified or subjectively determined; the training/evaluation data are not widely available.

**Reviewer Confidence:**

3: Pretty sure, but there's a chance I missed something. Although I have a good feel for this area in general, I did not carefully check the paper's details, e.g., the math, experimental design, or novelty.

---

> ### Author Rebuttal · Authors · 2023-08-28
>
> Thanks very much for taking the time to review this manuscript. We really appreciate all your generous comments and recognition of this paper! We will release the source code after the acceptance of this paper to facilitate reproducibility by readers.
>
> **Answer of Question#1:**
> &ensp;&ensp;Thanks for your insightful comment. In Table 2, the term 'w/o' is employed for ablation analysis, aiming to investigate the impact of each component of our framework on model performance. For instance, w/o Neutral word debiasing aims to evaluate the impact of the removal of neutral word debiasing in TFD on model performance. **Please refer to the analysis in Section 5.2.2.**

---

### Official Review · Reviewer_X5kt · 2023-08-11

**Soundness:** 4

**Excitement:**

4: Strong: This paper deepens the understanding of some phenomenon or lowers the barriers to an existing research direction.

**Paper Topic And Main Contributions:**

The paper introduces a framework for reducing various types of spurious correlations in models for Emotion Recognition in Conversational data (ERC). The framework is model-agnostic and uses counterfactual reasoning in a post-hoc manner (i.e. at prediction time), by applying masking to extract the model biases and obtain the counterfactual examples. The model predictions on these examples are then used to un-bias the original input predictions (by subtraction). The framework is "instantiated" and evaluated for reducing the effect of class imbalance on the classification performance (i.e. "label bias"), reducing the effect of the speaker bias and neutral words bias on the predictions.
The performance of the framework is compared to that of various specific ERC models (e.g. DialogueRNN, COSMIC, etc) as well as general-purpose PLMs (e.g. ChatGPT, Curie) - both from the classification performance point of view and that of improving fairness (the imbalance divergence metric is used for this).

**Questions For The Authors:**

A. How would/does the framework extend to models which do not explicitly encode the context of the utterance (x^hat)?

B. Can you argue why the imbalance divergence is a good metric for fairness? Why should the class labels be uniformly distributed to achieve fairness? (I am aware other works have used it, but I might be missing something, so please clarify, because I understand it as a metric for measuring label distribution skewness, not fairness)

**Reasons To Accept:**

Interesting approach to reducing imbalance effects and the effects of co-occurence statistics on the classification performance of ERC models.

Good evaluation work performed, comparisons with SoA methods.

**Reasons To Reject:**

I am not convinced about using imbalance divergence to measure fairness. Also, I would not use the word fairness in the context of the biases discussed in this paper, because we are affected by their effect on the performance of the classification models only, perhaps that should be the only metric used (?).

I found section 3.4 a little difficult to read. It was unclear which were notations introduced by the authors (e.g. the c's, are you masking all words in (9)?) and the causal intervention application. The examples/relation to the speaker/word biases helped.

**Reproducibility:**

4: Could mostly reproduce the results, but there may be some variation because of sample variance or minor variations in their interpretation of the protocol or method.

**Reviewer Confidence:**

3: Pretty sure, but there's a chance I missed something. Although I have a good feel for this area in general, I did not carefully check the paper's details, e.g., the math, experimental design, or novelty.

**Typos Grammar Style And Presentation Improvements:**

"Considering that except for generative approaches, most methods rely on using Roberta for utterance representation extraction." - needs rephrasing

"Our TFD may also make mistakes, as shown in Fig. 4, where certain samples of majority classes." - needs rephrasing

"the model simplistically predicts" - "... simply ..."

Some table captions are below the tables, others are above.

Table 7 is confusing. Does the first  row show the average perf, while the second analyzez only dialogues w/o emotion shift, and the third row dialogues only with emotion shift, but using the framework also? (i.e. it is not a different method, but different test samples).

---

> ### Author Rebuttal · Authors · 2023-08-28
>
> We would like to express our great appreciation for your valuable comments to improve the quality of this manuscript.
>
> **Answer of Reject#1:**
> &ensp;&ensp;Thanks for your insightful comment. According to (Sweeney and Najafian, 2019), the greater the prediction skewness of a trained model, the more it provides unfair opportunities across predefined categories. Consequently, the trained model becomes increasingly characterized by unfair discrimination. We thus follow previous work to use the metric – imbalance divergence – to evaluate whether a prediction is unfair. Based on this, to evaluate the speaker and neutral word biases of a trained model, we average the relative word (i.e. speaker and neutral word) imbalance over all the testing utterances. Similarly, to evaluate the label bias of a trained model, we average its relative label imbalance over all the testing utterances.
> &ensp;&ensp;Regarding whether the performance of the models should be the only metric used, we acknowledge model performance is a crucial metric. Nonetheless, we recognize that the significance of model fairness cannot be disregarded in real-world applications. Perhaps, we can emphasize model performance as the primary metric and then discuss the model imbalance divergence or unfairness as a secondary metric. We sincerely appreciate your suggestions and will carefully consider them in the revision.
>
> *Sweeney, C., & Najafian, M. (2019, July). A transparent framework for evaluating unintended demographic bias in word embeddings. In Proceedings of the 57th Annual Meeting of the Association for Computational Linguistics (pp. 1662-1667).*
>
> **Answer of Reject#2:**
> &ensp;&ensp;Thanks for your valuable suggestion. In the given utterance, *"CHANDLER says Act like a processor, people will think you're a processor. You're right."* with the emotion "peaceful". We want to prevent the model from being influenced by neutral words (such as "think", which often appears in utterances with the emotion "joyful"). Similarly, we also aim to avoid the influence of the speakers (such as CHANDLER, who frequently expresses neutral and joyful emotions). Therefore, we extract neutral word biases and speaker biases, in equation (9). Specifically, in causal intervention operation, we mask tokens that do not belong to the set of speakers (CHANDLER is the only speaker in this example) to predict outcomes with speaker bias. We can also obtain prediction outcomes with neutral word biases by masking tokens that do not belong to the set of neutral words, such as "right". We will detail it in the revision.
>
> **Answer of Question A:**
> &ensp;&ensp;Thanks for your insightful question. Bias extraction in our TFD operates at the feature level, ensuring its independence from the model structure. This is supported by outcomes in Tables 4 and 5, showing TFD applied to various ERC models that explicitly (e.g., KET, etc.) or implicitly (e.g., DAG-ERC, etc.) encode the context of the utterance.
>
> **Answer of Question B:**
> &ensp;&ensp;Thanks for your valuable question. According to (Sweeney and Najafian, 2019), the greater the prediction skewness of a trained model, the more it provides unfair opportunities across predefined categories. Consequently, the trained model becomes increasingly characterized by unfair discrimination. We thus follow previous work to use the metric – imbalance divergence – to evaluate whether a prediction is unfair. Based on this, to evaluate the speaker and neutral word biases of a trained model, we average the relative word (i.e. speaker and neutral word) imbalance over all the testing utterances. Similarly, to evaluate the label bias of a trained model, we average its relative label imbalance over all the testing utterances. We will detail it in the revision.
>
> *Sweeney, C., & Najafian, M. (2019, July). A transparent framework for evaluating unintended demographic bias in word embeddings. In Proceedings of the 57th Annual Meeting of the Association for Computational Linguistics (pp. 1662-1667).*
>
> **Answer of Typos Grammar Style And Presentation Improvements:**
> &ensp;&ensp;Thanks a lot for your valuable suggestions, we have corrected them in the revision and will avoid these mistakes in the future. Furthermore, Table 7 reports results indicating that the model fails to perform better on samples with emotional shifts compared to those without. We will detail it in the revision.

---

### Meta-Review · Area_Chair_Q9Eh · 2023-09-16

**Recommendation:** 5

**Metareview:**

This paper introduces an efficient debiasing method at the prediction time, evaluating it on the Emotion Recognition in Conversations (ERC) task. The reviewers are all excited about the originality and usefulness of this method, and the reported results in this paper. Some parts of the paper need further rewriting and improvement and the reviewers must work on that.

---

### Decision · Program_Chairs · 2023-10-07

**Decision:**

Accept-Main

**Comment:**

This paper introduces an efficient debiasing method at the prediction time, evaluating it on the Emotion Recognition in Conversations (ERC) task. The reviewers are all excited about the originality and usefulness of this method, and the reported results in this paper. Some parts of the paper need further rewriting and improvement and the reviewers must work on that.